# Cisplatin-Loaded M1 Macrophage-Derived Vesicles Have Anti-Cancer Activity in Osteosarcoma

**DOI:** 10.3390/cells14201616

**Published:** 2025-10-17

**Authors:** Namrata Anand, Joseph Robert McCorkle, David S. Schweer, Lan Li, Kristen S. Hill, Melissa A. Fath, Derek B. Allison, Christopher L. Richards, Jill M. Kolesar

**Affiliations:** 1Markey Cancer Center, College of Medicine, University of Kentucky, Lexington, KY 40536, USA; 2Department of Pharmacy Practice and Research, College of Pharmacy, University of Kentucky, Lexington, KY 40508, USA; 3Department of Obstetrics & Gynecology, College of Medicine, University of Kentucky, Lexington, KY 40508, USA; 4Department of Chemistry, College of Arts and Science, University of Kentucky, Lexington, KY 40506, USA; 5Department of Pharmaceutical Sciences and Experimental Therapeutics, College of Pharmacy, University of Iowa, Iowa City, IA 52242, USA; 6Department of Pathology & Laboratory Medicine, College of Medicine, University of Kentucky, Lexington, KY 40506, USA

**Keywords:** Osteosarcoma, M1-Macrophage-derived vesicles, M1 macrophages, Chemotherapy-loaded-MVs, cisplatin toxicity, manufactured cell-derived vesicles

## Abstract

Osteosarcoma (OS) is a relatively rare bone malignancy that primarily affects children and young adults and is associated with significant morbidity and mortality. Cisplatin is a mainstay of treatment, but its efficacy is limited by off-target toxicities. Immunotherapy is not effective due to a poor antigenic tumor microenvironment. Here, we address these challenges by using manufactured M1 macrophage-derived vesicles (MVs) loaded with cisplatin. Human blood and mouse RAW 264.7 M1 macrophages were used to prepare empty (E-MVs) and cisplatin-loaded MVs (C-MVs). Human OS cell lines were used in vitro and in a tibia xenograft mouse model to evaluate the anti-cancer and immune-stimulating abilities of MVs. C-MVs had lower IC50s but equivalent DNA damage in OS cell lines when compared with free cisplatin. E-MVs and C-MVs were observed to accumulate in the tumor in OS tumor-bearing mice. C-MVs significantly reduced tumor burden and prolonged survival in a mouse model of OS. Animals dosed with free cisplatin experienced weight loss and renal and hepatic toxicity, while equivalent doses of C-MVs did not cause these effects. In addition, both E-MVs and C-MVs showed immunomodulation of the tumor microenvironment with a significant increase in the M1/M2 macrophages ratio (7-fold and 22-fold, respectively) and increased levels of TNF-α in serum (1.8-fold and 2.1-fold, respectively) compared to control mice. Collectively, these experiments support further development of C-MVs for the treatment of OS.

## 1. Introduction

Osteosarcoma (OS) is a rare primary malignant bone tumor occurring primarily in children, with 3770 new cases estimated for 2025 [1]. Historically, osteosarcoma was managed with surgery alone, and approximately 80% of patients developed metastatic disease. Although the introduction of adjuvant chemotherapy has improved 5-year survival from approximately 20% to 60%, metastasis continues to be the most important factor driving poor survival outcomes. Currently, OS treatment includes neoadjuvant chemotherapy followed by surgical resection and adjuvant therapy. While there is no consensus on a standard chemotherapy regimen, most regimens include cisplatin and doxorubicin with or without high-dose methotrexate [2,3].

Although primarily immune checkpoint inhibitors (ICIs) have revolutionized the treatment of many cancers, ICIs have been relatively ineffective for OS [4,5,6]. The lack of activity of ICIs is associated with a “cold” tumor microenvironment, which is characterized by the presence of M2 macrophages and other cells that secrete immunosuppressive cytokines [7]. In the tumor microenvironment, factors released by tumor cells can cause tumor-associated macrophages (TAMs) to develop an M2-like, anti-inflammatory phenotype. In general, M2 macrophages are associated with increased metastasis, angiogenesis, chemoresistance, and poor prognosis of many tumors. In primary OS, M2 makes up a large proportion of the TAMs and is associated with enhanced metastatic potential and worse survival outcomes [8]. A therapy that reprograms M2 tumor-associated macrophages into the pro-inflammatory M1 subtype could transform immunologically “cold” tumors into “hot” tumors—making them more responsive to immune-based treatments. This approach represents a potentially paradigm-shifting advance in osteosarcoma therapy.

Extracellular vesicles (EVs) are naturally occurring nanoscale lipid membrane structures released by nearly all cell types that function in cell-to-cell communication. EVs have a lipid bilayer, lack a nucleus, and serve as carriers of proteins, nucleic acids, and other substances from their origin cells, which they deliver to recipient cells [9]. EVs are under investigation as drug delivery vesicles and microenvironment modulators for a number of diseases; however, their low production rate in unaltered cell cultures and lack of batch reproducibility are current barriers to clinical development.

Manufactured cell-derived vesicles (MCDVs) are EV-like vesicles produced by cell disruption strategies. A novel method for generating via fragmentation of cell membranes was recently reported [10]. These MCDVs retain properties of naturally occurring EVs, but importantly, can be produced at scale [10,11,12]. Macrophage-manufactured vesicles (MVs) are a type of MCDV, made from human and mouse M1 macrophages. Previous studies demonstrate that cell-derived vesicles exhibit similar properties to exosomes in that they specifically target the cell from which they originate [12]. As macrophages are the most abundant immune system cell within the tumor microenvironment, MVs formulated from macrophages have inherent targeting properties to most tumors. Previous animal data demonstrate precise localization of mouse MVs to cancer tumor xenografts in mice [12,13]. In addition, (MCDVs) can be engineered to carry chemotherapeutic agents, enabling targeted delivery directly to cancer cells [12,13]. This approach leverages the natural ability of MCDVs to interact with specific cell types through surface markers, thereby enhancing drug accumulation at the tumor site while minimizing off-target toxicity. By exploiting this cell-specific delivery mechanism, EV-based chemotherapy holds promise for improving therapeutic efficacy and reducing systemic side effects commonly associated with conventional cancer treatments.

The purpose of this study was to characterize MVs generated from both primary human monocytes and the RAW 264.7 mouse monocytic cell line, and to better understand cellular internalization and targeting. Anti-cancer efficacy and safety of empty (E-MVs) and cisplatin-loaded MVs (C-MVs) were assessed in vitro and in a mouse orthotopic xenograft model of OS (see Graphical Abstract).

## 2. Materials and Methods

### 2.1. Cell Lines and Culture Media

Human OS parental cells HOS (ATCC: CRL-1543), its derivative cell line 143B (CRL-8303), and mouse macrophage cell line RAW 264.7 (TIB 71) were obtained from ATCC, Manassas, VA, USA. HOS cells were maintained in Eagle’s Minimum Essential Medium (ATCC, Manassas, VA, USA #30-2003) supplemented with 10% fetal bovine serum (FBS) (Sigma-Aldrich, Saint Louis, MO, USA #F0926). 143B cells were maintained in Minimum Essential Medium in Earle’s Balanced Salts (Sigma-Aldrich, Saint Louis, MO, USA #51412C) supplemented with 10% FBS, 1% penicillin-streptomycin (Gibco, Brooklyn, New York, NY, USA #15140-122), and 0.015 mg/mL 5-bromo-2′-deoxyuridine (BrdU) (Sigma-Aldrich, Saint Louis, MO, USA #B5002). RAW 264.7 cells were maintained in DMEM (Sigma-Aldrich, Saint Louis, MO, USA #D0822) media supplemented with 10% FBS and 1% penicillin-streptomycin.

### 2.2. Isolation of Human Peripheral Blood Mononuclear Cells (PBMCs) and Differentiation of M0 to M1 Macrophages

Human blood was obtained from the University of Kentucky Blood Center (IRB-58723 Approval for Exemption certification, 12 April 2020) and processed for the isolation of PBMCs through density gradient centrifugation using Ficoll-Paque (GE Healthcare, Tampa, FL, USA#17-5446-52). A total volume of 200–250 mL of blood product was isolated and pooled from 5 to 6 buffy coat bags for a single vesicle preparation. The Easy Sep human monocyte enrichment kit (Stemcell Technologies, Vancouver, BC, Canada, #19058) was used to isolate monocyte/M0 macrophages from human PBMCs using negative selection. Isolated M0 macrophages were cultured in RPMI 1640 media (Sigma-Aldrich, Saint Louis, MO, USA #R8758), supplemented with 10% heat-inactivated FBS, 1% penicillin-streptomycin, and recombinant human macrophage colony-stimulating factor (M-CSF) at 50 ng/mL (PeproTech, Oil City, PA, USA #300-03-5µG). Cells were maintained in a humidified incubator at 37 °C with 5% CO_2_ for 6 days, and the M-CSF-containing media was changed every 48 h. After 6 days, M-CSF-containing media were removed, and cells were stimulated to M1 macrophages by adding 20 ng/mL lipopolysaccharide (LPS) (Invitrogen, Carlsbad, CA, USA #tlrl-eblps) and 20 ng/mL recombinant human interferon-γ (IFN-γ) (PeproTech, Oil City, PA, USA #300-02-500UG) for 24 h.

### 2.3. RAW 264.7 Based M0 and M1 Macrophages Preparation

RAW 264.7 macrophage cells, at an approximate count of ~3 × 10^9^, were used to prepare the murine (m) E-MVs and mC-MVs for in vivo experiments in mice. RAW 264.7 cells were stimulated with IFN-γ and LPS to polarize to the M1 phenotype as described previously for the human PBMC-derived macrophages.

### 2.4. Vesicles Generation

MVs were generated from M1 macrophages obtained from human monocytes and RAW 264.7 mouse cell lines through nitrogen (N2) cavitation as described previously [8,13]. Briefly, M1 macrophages were scraped from the cell culture flask using ice-cold phosphate-buffered saline (PBS) (Sigma-Aldrich, Saint Louis, MO, USA #P2272) with Pierce protease inhibitor (PI) mini-tablets (Thermo Scientific, Madison, WI, USA #A32953). Cells were centrifuged at 300× *g* at room temperature (RT) to remove any remaining cytokines and washed with PBS containing PI. C-MVs or E-MVs were prepared as described in this section. Approximately 100 million human M1 macrophages were used to perform N_2_ cavitation. To prepare C-MVs, an 8 mM solution of cisplatin (Tocris Bioscience, Bristol, UK #2251) in PBS was prepared and kept at 37 °C for 3–4 h to dissolve the cisplatin. Cells were scraped from the flask, pelleted down, and resuspended in 8 mL cisplatin solution. E-MVs were prepared by resuspending macrophage cell pellets in 10 mL of PBS. The cell suspensions prepared for either C-MVs or E-MVs were added to a pre-chilled pressurized chamber (Parr Instruments Company, Moline, IL, USA), which was then pressurized with 250 psi N_2_ gas for 5 min at 4 °C to allow the cavitation to occur. The cell homogenate was collected from the chamber, and vesicles were purified from cellular debris by three centrifugation steps at increasing speeds. The first centrifugation was at 4000× *g* for 20 min at 4 °C. The supernatant containing the vesicles was collected, and the cellular debris within the pellet was discarded. The supernatant was further centrifuged at 10,000× *g* for 20 min at 4 °C to concentrate the vesicles. Again, the supernatant was collected and ultracentrifuged at 100,000× *g* for 1 h at 4 °C was used to pellet the vesicles. The final vesicle pellet was resuspended in 1 mL of PBS.

### 2.5. Nanoparticle Tracking Analysis (NTA)

NTA was used to count the E-MVs and C-MVs using Nanosight NS300 (Malvern Panalytical, Malvern, UK). Samples were measured using constant camera and detection settings for all acquired images and videos. Each sample was recorded for 1 min, and a total of five repetitions with more than 200 tracks per video. The particle count analysis was performed using NTA version 3.4 software.

### 2.6. Characterization of Vesicles Through Scanning Electron Microscopy (SEM)

RAW 264.7 E-MVs were fixed in 2% formaldehyde (Thermo Fischer Scientific, Madison, WI, USA #J61899.AK) for 1 h and washed with PBS three times for 10 min at RT, as described previously [13]. MVs were then further dehydrated using a series of ethanol (Sigma Aldrich, Saint Louis, MO, USA #1085430250) washes from 50%, 60%, 70%, 80%, 90%, 95%, and 100% for 10 min each. MVs were finally resuspended in 200-proof ethanol until imaged using SEM. MVs were briefly sonicated, and then a drop of the sample was transferred into a critical point dryer (EM CPD 300, Leica Microsystems). The sample was then metalized by sputter-coating 5 nm platinum (EM ACE 600, Leica Microsystems, Buffalo Grove, IL, USA) to enhance surface electrical conductivity. MVs were imaged using a field emission electron microscope, FE-SEM, Helios Nanolab 660 (Thermo Scientific, Madison, WI, USA).

### 2.7. Transmission Electron Microscopy

Human monocyte-derived E-MVs were tracked inside OS cells using immunogold silver enhancement staining (IGSS) and imaged using transmission electron microscopy (TEM). 143B cells were incubated with MVs for 24 h at 37 °C. After incubation, cells were washed with PBS and fixed with 4% paraformaldehyde for 40 min at RT. MVs were located using the CD86 antibody (Abcam, Cambridge, MA, USA #A243887), which acts as an M1 marker and primary antibody, and gold nanoparticles (Nanoprobes, Yaphank, NY, USA #2003) were used as a secondary stain. The immune gold silver enhancement staining (IGSS) (Nanoprobes, Yaphank, NY, USA #2012) method was performed, as described previously, to visualize the vesicles [14].

### 2.8. LC-MS for Quantification of Cisplatin in C-MVs

Quantification of cisplatin was performed using the method described by Shaik et al. [15]. A 10 mg/mL stock solution of cisplatin was prepared in dimethyl sulfoxide (DMSO) (Invitrogen, Carlsbad, CA, USA #D12345) and diluted in 50% methanol (Sigma-Aldrich, Saint Louis, MO, USA MX0486) in water to make a series of standard solutions (5, 10, 20, 50, 100, 200, 500 ng/mL) and quality controls (5, 25, 100, 500 ng/mL). To liberate cisplatin from C-MVs, vesicle samples were combined 1:1 with methanol, vortex mixed, and subjected to 5 successive freeze–thaw cycles using liquid nitrogen. 100 µL of each sample, standard, and quality control was combined in fresh polypropylene microcentrifuge tubes with 20 µL of 5 µg/mL transpalladin (trans-Diamminedichloropalladium (II)) (ThermoFisher Scientific, Madison, WI, USA #011037-02) and 20 µL of 1% diethyldithiocarbamic acid, sodium salt trihydrate (DTC) (Enzo Life Sciences, Farmingdale, NY, USA #ALX-400-003) in 0.1 N sodium hydroxide. The tubes were vortexed for 2 min, then incubated for 30 min at 4 °C to form a DDTC complex with platinum from cisplatin (Pt-DDTC) and palladium from transpalladin (Pd-DDTC). 300 µL of acetonitrile (Sigma-Aldrich, Saint Louis, MO, USA #AX0156) containing 50 ng/mL 8-cyclopentyl-1,3-dipropylxanthine (DPCPX) (Sigma-Aldrich, Saint Louis, MO, USA #C101) was added, then vortexed for 2 min, and centrifuged at 15,000× *g* for 5 min. Supernatant (250 µL) was transferred to 96-well polypropylene plates and dried under a stream of N_2_ gas. The dried sample was resuspended in 100 μL of a 4:1 mixture of 0.1% formic acid (Agilent Technologies, Ankeny, IA, USA #G2453-85060) in water/acetonitrile and placed into the autosampler.

Samples were analyzed using a 1260 Infinity II LC system (Agilent Technologies, Ankeny, IA, USA) interfaced with a Ultivo triple quadrupole mass spectrometer (Agilent Technologies, Ankeny, IA, USA). Liquid chromatography was performed using an Agilent Infinity Lab Poroshell HPH-C18 column (100 mm × 2.1 mm, 2.7 μm). The mobile phase consisted of 0.1% formic acid in water (A) and acetonitrile (B) delivered as a linear gradient as follows: 0–0.5 min, 5% B; 0.5–1 min, 75% B; 1–1.5 min, 90% B; 1.5–2.5 min, 95% B; 2.5–4 min, 95% B; 4–5 min, 5% B and a 10 min equilibration before injection of the next sample. The flow rate was 0.35 mL/min. Electrospray ionization was operated in positive mode (ESI+) with nitrogen as both curtain and collision gas. The Ultivo instrument parameters were as follows: drying gas temperature, 30 °C; drying gas flow, 10 L/minute; sheath gas temperature, 25 °C; sheath gas flow, 11.5 L/minute; and nebulizer pressure, 45 psi. MRM used to detect the transitions of Pt-DDTC at *m*/*z* 491 → 88 (fragmentor voltage 240 V; collision energy 41 V), Pd-DDTC at *m*/*z* 403 → 116 (fragmentor voltage 190 V; collision energy 21 V), and DPCPX at *m*/*z* 305.2 → F178 (fragmentor voltage 98 V; collision energy 37 V). The data were analyzed using MassHunter Workstation (Agilent Technologies, Ankeny, IA, USA, version 1.1) software.

### 2.9. Drug Response Assay on HOS and 143B

HOS and 143B cells were plated at 5000 cells/well onto a white-bottom 96-well cell culture plate and incubated for 24 h. Six different concentrations of serially, threefold diluted free cisplatin were used from 0.41 µM to 100 µM. Human (h) PBMC-derived hC-MVs were used at six different serial, threefold dilutions from 0.082% to 20%. The cisplatin concentration in the hC-MVs was determined by LC-MS to range from 0.3 ± 0.1 µM (0.083%) to 80 ± 33 µM (20%). Control cells were incubated in complete media without any added drugs. Cells were incubated for 96 h, and cell viability was assessed using Cell Titer-Glo 2.0 (Promega, Madison, WI, USA #G9243). Data were normalized to control cells and expressed as a percentage of cell viability. The assay was performed in 3 independent replicates, with each dose performed in duplicate (n = 3). Dose–response curves were generated using GraphPad Prism (version 5.1) and were used to calculate IC50 values.

### 2.10. DNA Damage Assay

HOS and 143B cells were seeded into a walled, clear-bottom 96-well cell culture plate (Thermo Scientific, Madison, WI, USA #165305) at 10,000 cells per well and allowed to adhere in a humidified CO_2_ incubator at 37 °C. After 24 h incubation, old media was replaced with fresh media having free cisplatin or hC-MVs at concentrations near the IC50 values for each cell line. Complete media, without any drug or vesicles, was used as a negative control. Cells were incubated for 24 h, followed by fixation with 4% paraformaldehyde (Thermo Scientific, Madison, WI, USA) in PBS at RT for 15 min. Cells were then permeabilized by incubating with 0.25% Triton X-100 (Alfa Aesar, Haverhill, MA, USA #A16046.AE) in PBS for 15 min and then blocked by incubation with 0.1% BSA (Proliant Biologicals, Ankeny, IA, USA #10842-662) in PBS for 1 h. Double-strand DNA breaks in cells were detected by a fluorescent antibody against phosphorylated histone H2AX (pH2AX) and nuclei were counter-stained using Hoechst 33342 (HCS DNA Damage Kit, Invitrogen, Carlsbad, CC, USA #H10292). Cells were imaged using the Cell Insight CX7 High Content Analysis Platform (Thermo Scientific, Madison, WI, USA), and quantification of nuclear pH2AX signal was performed using the HCS studio software (Thermo Scientific, Madison, WI, USA). Channel 1 defined the nucleus with Hoechst 33342 (using the segmentation tool) as an object with Hoechst/XF93 filters. Channel 3 determined the fluorescence intensity of the pH2AX signal using TRITC/XF93 filters in the nucleus outlined in channel 1. The fold change in pH2AX intensity was calculated by normalizingto matched control cells, and graphs were generated using GraphPad Prism (version 5.01).

### 2.11. In Vivo Mouse Experiment

Animal studies were approved by the University of Kentucky Institutional Animal Care and Use Committee (Approval #2017-2674). A total of 5 × 10^5^ luciferase-labeled 143B cells were orthotopically injected into one tibia of 6–7-week-old female and male outbred homozygous nude mice (Jackson Laboratory, Bar Harbor, ME, USA #007850). A total of 32 mice were used in the study, with 8 mice in each group. The number of mice per group was determined a priori based on previously published xenograft growth experiments. Confounders of the order of treatments and the animal cage effect were not controlled for. Mice were treated with meloxicam analgesic at 5 mg/kg for 3 days after tumor cell injection. Tumor volume was assessed weekly using whole body bioluminescence imaging following intraperitoneal injection (IP) of 150 mg/kg of IVISbrite D-luciferin potassium salt (PerkinElmer, Waltham, MA, USA #760504) on Lago X animal imager (Spectral Instruments Imaging, Tucson, AZ, USA). The bioluminescence was not conducted by a blinded investigator; however, instrument settings were determined a priori and remained constant throughout the measurements. The bioluminescence values were converted to log values and baseline corrected. Animals were randomly assigned to four treatment groups when total body luminescence was between 2.0 and 7.5 × 10^7^ photons/second: free cisplatin, mC-MVs, mE-MVs, and control (PBS). The concentration of cisplatin in mC-MVs was measured using LC-MS, and the amount of free cisplatin used for each treatment was matched to the amount of cisplatin in the mC-MVs treatment each week. The mice were treated IP once per week until they met criteria for euthanasia or for a total of 12 weeks. Mice not previously euthanized were monitored without treatment for 2 weeks post-end of treatment. Endpoint criteria included a 15% decrease in body weight from the start of treatment or a tumor burden that precluded movement. At the endpoint, mice were humanely euthanized, blood was collected using cardiac puncture, and the tumor, liver, spleen, kidney, and lungs were harvested.

### 2.12. MVs Labeling and Localization

Murine (m) RAW 264.7 cell-generated mMVs were resuspended in 2 mL of 250 mM sucrose buffer (VWR, Radnor, PA, USA #76177-754) in 10 mM HEPES (VWR, Radnor, PA, USA #97061-770), pH 7.5. Lipophilic DiR near-infrared fluorescent dye (Thermo Fisher Scientific, Madison, WI, USA #D12731) was diluted to a final concentration of 2 mM in sucrose buffer and incubated with mMVs for 30 min at 37 °C, after which, the vesicle solution was carefully layered with a 50% and 10% OptiPrep density gradient medium (Millipore Sigma, Madison, WI, USA). The sample was ultracentrifuged at 112,000× *g* for 1 h at 4 °C to separate labeled vesicles. Labeled vesicles were collected using a peristaltic pump in a 1.5 mL Eppendorf tube and were further purified using size exclusion PD Miditrap columns (Cytiva, Marlborough, MA, USA) to remove any free dye. One 6-week-old nude mouse bearing a human 143B tibia xenograft was used to investigate the localization of the mE-MVs. A total of 200 µL of dye-labeled E-MVs was injected intraperitoneally into the mouse and was imaged 48 h post-injection using Lago X animal imager (Spectral Instruments, Tucson, AZ, USA) at a fluorescence excitation of 710 nm and emission of 770 nm for 10 s to determine the localization of the mE-MVs in vivo. After imaging, a necropsy was performed, and the tumor, liver, lungs, and spleen were harvested and imaged in the same way as the whole mouse imaging. Fluorescent images were analyzed with Aura version 4.0.8 software.

### 2.13. Mouse Serum Profiles

Diagnostic chemistry profiling was performed on serum samples from all mice at the time they met criteria for euthanasia. Blood samples were collected via cardiac puncture under isoflurane anesthesia into Eppendorf tubes. The serum was separated from the blood within 30 min of collection by centrifugation at 2000× *g* for 15 min at RT. 100 µL of serum sample was used to perform serum chemistry for alkaline phosphatase (ALP), alanine aminotransferase (ALT), total albumin, creatinine, blood urea nitrogen (BUN), bilirubin, calcium, potassium, sodium, glucose, and phosphate using an automated analyzer, Abaxis VetScan version S2.

TNF-α was evaluated in mouse serum using a Quantikine ELISA kit (Bio-Techne, Minneapolis, MN, USA #MTA00B) according to the kit’s instructions. One mouse each in control and free cisplatin groups had TNF-α values below the level of quantification, and therefore, values were extrapolated using the equation of the standard curve [16].

### 2.14. Histopathology of Mice Organs

Tissues, including the xenograft tumor on the tibia, liver, and kidney, were collected once the mice reached humane endpoints (described above) or after a total of 14 weeks. All tissues were fixed in 10% formalin (VWR, Radnor, PA, USA #100496-502) for 48 h and then transferred to 70% ethanol, followed by a series of dehydration before embedding in paraffin. Tissue sections were cut at 4–5 µm thickness and stained with hematoxylin and eosin (H&E) for histopathologic analysis by a blinded pathologist.

### 2.15. Immunohistochemistry for M1 and M2 Macrophages

Dewaxing of the parafilm-embedded sections was performed at 58 °C oven for 2 h. After dewaxing, the slides were cooled and passed through a series of 100% xylene (Sigma-Aldrich) and serially dehydrated from 100% to 70% ethanol. After dehydration, antigen retrieval was performed using a high-pH Tris base buffer (pH 9.0) (Abcam, Cambridge, MA, USA) for 15 minutes at 98 °C. The tumor sections were stained with anti-rabbit monoclonal antibody for CD86 (M1 macrophage, Abcam, Cambridge, MA, USA #A243887) and anti-rat monoclonal Ab for CD163 (M2 macrophage, Abcam, Cambridge, MA, USA #289979). The M1 and M2 macrophages were visualized by using anti-rabbit Alexa Fluor 488 (green fluorescence; Themo Fischer Scientific, Madison, WI, USA #A-11008) and anti-rat Alexa Fluor 594 fluorochromes (red fluorescence; Thermo Fischer Scientific, Madison, WI, USA #A11007) for M1 and M2 macrophages, respectively. The nuclei were visualized by DAPI (Thermo Fischer Scientific, Madison, WI, USA #62248) fluorescent staining. Slides were examined using a confocal microscope (Leica Biosystems, Buffalo Grove, IL, USA). Five similarly sized regions of interest (ROI) were randomly selected in the tumor slide, and total CD86 (M1) or CD163 (M2) positive cells were counted, and M1/M2 ratios were plotted for all the mice.

### 2.16. Statistical Analyses

All in vitro experiments included at least two technical replicates and were performed at least three times independently. Non-linear regression analysis was used to plot the dose–response curve and the log (inhibitor) vs. response curve. A variable slope equation was used to calculate the cell viability assays and IC50 calculations. The student’s unpaired t-test was used to analyze the difference between the IC50 values and the DNA damage assay between free cisplatin and C-MVs in cell lines. For mouse bioluminescence, two-way ANOVA was used. For analyzing the weight, serum chemistry, and IHC results, one-way ANOVA with post hoc test specified in the figure legend. For analyzing the Kaplan–Meier survival curves, a log-rank test was used.

## 3. Results

### 3.1. Anti-Cancer and DNA Damage Potency of Free Cisplatin vs. C-Mvs

To compare the anti-cancer effects of hC-MVs to cisplatin, two human OS cell lines, HOS and 143B, were treated with six serial dilutions of hC-MVs, hE-MVs, and free cisplatin, and a toxicity assay was performed. hC-MVs had a significantly lower IC50 when compared to free cisplatin in both HOS and 143B cells. The mean free cisplatin IC50 value in HOS cells was 4.0 ± 0.2 µM compared to 1.4 ± 0.7 µM for hC-MVs (n = 3, *p* < 0.05), and for 143B cells, the mean IC50 for free cisplatin was 3.7 ± 0.3 µM compared to hC-MVs 1.1 ± 0.5 µM (n = 3, *p* < 0.05) (Figure 1A). These data suggest hC-MVs are more potent than free cisplatin. hE-MVs, without the cisplatin payload, also showed inhibition of cell proliferation in osteosarcoma cell line (*p* = 0.03 in HOS cells, *p* = 0.06 in 143B) for the highest concentration tested (Appendix A).

Cisplatin exerts its cytotoxicity primarily by forming DNA adducts and cross-links that stall replication forks and lead to double-strand breaks, which can be detected by the histone pH2AX [17]. Measuring nuclear pH2AX, therefore, provides a sensitive readout of cisplatin-induced DNA damage and is appropriate for comparing the genotoxic effects of free drug versus drug delivered in vesicles. To assess the ability of hC-MVs to induce DNA damage, OS cells were treated at the approximate IC50 concentrations of cisplatin and hC-MVs (3.7 µM and 1.5 µM, respectively) for 72 h. Both HOS and 143B cell lines showed similar levels of DNA damage when treated with hC-MVs and cisplatin, as demonstrated by the mean fold change in nuclear pH2AX intensity. HOS cells demonstrated a 4.5 ± 0.8 mean fold increase in pH2AX intensity with free cisplatin treatment over control-treated, and a 4.6 ± 0.6-fold increase for hC-MVs over controls. 143B cells had 5.9 ± 1.2-mean fold increase in pH2AX intensity with free cisplatin treatments, and 4.4 ± 0.5 with hC-MVs treatment (Figure 1B–D, *p* = NS). These results suggest that lower concentrations of cisplatin delivered by hC-MVs resulted in similar DNA damage to free cisplatin.

### 3.2. Mouse RAW 264.7 M1 Derived MVs Localization

To evaluate the timing, targeting, as well as off-target distribution of M1-MVs when delivered intraperitoneally (IP), information essential for developing MV-based OS therapies, E-MVs were prepared from RAW 264.7 by nitrogen cavitation and characterized by SEM. Similarly to our previously published results evaluating human PBMC hE-MVs, RAW 264.7 murine (m) derived mE-MVs have a smooth, round morphology and are approximately 100–200 nm in diameter (Appendix A). Macroscopic localization of mE-MVs was evaluated in vivo in an OS orthotopic mouse model. DiR dye-labeled mE-MVs were injected IP into a nude mouse bearing a 143B tibia xenograft. Approximately 48 h after the injection, mE-MVs localize near the intratibial tumor as demonstrated by the high fluorescence intensity (Figure 2A). Upon necropsy of the animal, the fluorescence signal intensity from the DiR dye labeled mE-MVs was identified in the primary tibia tumor and localized in the liver, but not in the spleen, lungs, or kidney (Figure 2B). Interestingly, fluorescence was also observed in the liver.

### 3.3. mC-MVs Treatment Resulted in Tumor Growth Inhibition and Prolonged Survival in Mice

We evaluated the anti-cancer efficacy of murine macrophage RAW 264.7 cell-derived mE-MVs, mC-MVs, free cisplatin, and control in a 143B-luc orthotopic model of osteosarcoma. Thirty-two total outbred homozygous nude male and female mice were injected with 5.0 × 10^5^ 143B-luc cells in the right tibia. Mice were followed with weekly bioluminescence imaging to assess tumor growth, and a minimal threshold radiance of 2 × 10^7^ was used to distinguish tumor establishment and inclusion for treatment. At randomization, tumor size was not significantly different between groups (Appendix A, *p* = 0.90). Mice were assigned to one of four groups of 8 mice each, and treatment was given IP weekly until the mice met criteria for euthanasia or the planned end of treatment of 12 weeks. Cisplatin concentration in mC-MVs was assessed by LC-MS, and the free cisplatin dose was matched to the mC-MVs dose, with a median cisplatin dose of 4.2 ± 0.6 mg/kg. The particle counts of mC-MVs and mE-MVs were matched weekly with an average particle count of 6.56 × 10^11^ ± 9.04 × 10^10^ delivered per dose.

At six and seven weeks of treatment, a significant difference was observed in average tumor size between control mice and mC-MV-treated mice as measured by bioluminescence imaging (Figure 3A, * *p* < 0.05). At week eight, the majority of control mice had died or been humanely euthanized; however, the majority of the mC-MVs-treated mice had slow tumor growth until treatment was stopped (Figure 3A). There was no significant difference in mean tumor size between any other treatment groups.

The pre-determined endpoint criterion included a 15% decrease in body weight from the start of the treatment, or tumor burden precluding movement, at which time mice were humanely euthanized. Most control mice showed excessive tumor burden precluding movement starting by week 6 of treatment, and all were humanely euthanized by day 74, with a median survival of 56.5 (range 48–74) days. Of the mice treated with free cisplatin, three were euthanized due to weight loss, and the rest were euthanized due to excessive tumor burden, with a median survival of 59 days (range of 42–74). Of the mice in the mE-MVs treatment group, one of eight survived to day 98 (the predetermined end of the experiment), and the median survival was 65 days (range, 43–98 days). Two of the eight mice treated with mC-MVs survived for 98 days, and the median survival time was 86 days (range, 43–98 days). Survival of mice treated with mC-MVs was significantly improved compared to the control group (Figure 3B, *p* = 0.014).

It has previously been established that 143B cells readily form lung metastasis [18]. Therefore, upon euthanasia, the lungs were removed and metastasis counted. Six out of eight mice (75%) developed metastasis in the control group, while only 50% of the mice in each of the treatment groups had metastasis. The mean number of metastatic nodules was highest in the control group (13.8 ± 6.1), followed by the mC-MVs treated group (7.3 ± 4.4, *p* = 0.25 vs. control), then in the mE-MVs treated group (3.0 ± 1.4, *p* = 0.059 vs. control). The control group was significantly different from the free cisplatin group, 2.37 ± 1.0 (Figure 3C, *p* = 0.047). However, while there were no significant differences between the control and mE-MV or mC-MV, it should be noted that metastasis was assessed at death or euthanasia, and these mice had significantly longer times to develop metastasis when compared to the controls.

### 3.4. mC-MVs Are Less Toxic than Free Cisplatin Treatment at Equivalent Doses

Mouse body weight is often used as an indicator of a mouse’s health. Each mouse in the control and mE-MVs treatment groups had increased body weight during treatment. Mice treated with mC-MVs increased in weight but at a significantly slower rate than control mice (Figure 4A, *p* = 0.014). In the free cisplatin treatment group, three mice lost ~15% body weight by week 9 of treatment, which was a pre-determined endpoint for euthanasia, and none of the mice gained weight. At week 10 of treatment, the mean normalized weight of mice treated with free cisplatin was significantly less than mice in either the control or mC-MVs groups (Figure 4A, * *p* < 0.001 and *p* = 0.03, respectively).

Common side effects of cisplatin therapy in humans are hepatic and renal toxicity; therefore, blood drawn from all 32 mice, via cardiac puncture, at euthanasia, was evaluated for clinical chemistry endpoints to understand potential toxicities. All mice demonstrated normal serum creatinine levels, which are generally used as a marker of kidney function (Figure 4C). Three mice treated with free cisplatin had abnormally high BUN levels, and the average BUN level in this group was significantly higher than the average BUN in the mE-MVs or mC-MVs groups (Figure 4B, *p* < 0.05). Gross liver function is often measured by serum alkaline phosphatase (ALP) and alanine aminotransferase (ALT) levels. Mice treated with free cisplatin had an average ALP level that was significantly higher than mice treated with mE-MVs (Figure 4D, *p* < 0.05). The mean ALT serum levels did not differ between treatment groups (Figure 4E). Importantly, the ratio of albumin to globulin is considered a more sensitive measurement of liver and kidney function, as well as a marker for general health. This ratio was above the normal limit in 6 of 8 mice treated with free cisplatin, and the mean was significantly more than the mean of any other group, indicating that this treatment was toxic to the mice (Figure 4F, *p* < 0.05). Other serum electrolytes, including calcium, potassium, and sodium, were significantly higher in free cisplatin mice when compared with controls, but mice treated with mE-MVs and mC-MVs did not have electrolyte abnormalities (Appendix A).

### 3.5. mC-MVs and mE-MVs Did Not Cause Pathological Changes in the Liver or Kidneys of Treated Mice

Although serum creatinine and BUN are commonly used as markers for acute kidney injury, they are suboptimal as they are dependent on non-renal factors, and they are not sensitive to minor changes in kidney function. Therefore, to more closely evaluate the safety of treatments, the kidneys were removed from all mice, and H&E staining was performed and evaluated by a blinded pathologist. Kidney tissue from free cisplatin-treated mice showed an increased vacuolization in renal epithelial cells, which corresponds to a loss of mitochondria (Figure 5B), which is a well-documented feature of drug-induced renal injury. Control, mE-MVs, and mC-MVs-treated mice showed no vacuolization of renal cells and showed normal dense cytoplasm of renal epithelial cells (Figure 5A,C,D). Likewise, the livers of all the treated animals were examined by a blinded pathologist. H&E staining of the liver in control, mE-MVs, and mC-MVs treated mice demonstrated intact hepatocytes throughout the liver and no signs of necrosis (Figure 5E,G,H), while mice treated with free cisplatin showed hepatocyte necrosis in many areas of the liver (Figure 5F). Taken together, these data clearly indicate that mC-MVs treatment is less toxic than similar free cisplatin concentrations.

### 3.6. MV Treatment Results in a More Immunogenic Phenotype as Demonstrated by an Increased M1/M2 Ratio of TAMs and an Increase in Serum TNF-α

Immunohistochemical analysis of M1 and M2 macrophages was performed on the OS tibia tumor harvested and sectioned from mice. Tumors (Control n = 8, Free cisplatin n = 5, mE-MVs n = 8, mC-MVs n = 6) were stained with the M1 marker for CD86 antibody (green), the M2 marker for CD163 (red), and the nuclei were visualized with DAPI (blue) (Figure 6A). Three mice from the free cisplatin group and two mice from the mC-MVs had tumors too small to visualize and were excluded from the assessment. M1 and M2 macrophages were counted and summed in five randomly selected regions of interest (ROI) of each tumor. M1/M2 ratios were calculated over all 5 ROI. The highest M1/M2 ratio (average ± SEM) was detected in mC-MVs mice (21.9 ± 10.2), followed by mE-MVs (7.4 ± 3.6). Control mice had a significantly lower mean M1/M2 ratio of (1.1 ± 0.5) than mE-MVs or mC-MVs (Figure 6B, *p* < 0.05). The mean M1/M2 ratio in free cisplatin mice (1.6 ± 0.6) was not significantly different from control mice (Figure 6B). These results indicate that the outcome of treatment with M1-MVs is the recruitment of M1 macrophages and/or the repolarization of M2 macrophages to an M1 phenotype, with the consequence of creating a more inflammatory tumor microenvironment.

TNF-α is one of the primary cytokines secreted by activated M1 macrophages. Therefore, to provide insight into systemic immune activation and the inflammatory response triggered by MVs treatment, we quantified TNF-α in the serum of mice treated with control (n = 8), mE-MVs (n = 8), mC-MVs (n = 8), and free cisplatin (n = 8). As serum was drawn at euthanasia, two mice in the mC-MV group and one mouse in the mE-MV group had serum drawn two weeks following the last treatment, and serum from one mouse each from the control group and free cisplatin had values below the limit of quantification and are represented with extrapolated values (Figure 6C). The highest levels of TNF-α (average ± SEM) were found in the mC-MVs group with a mean of 35.5 ± 3.7 pg/mL, followed by mE-MVs with a mean of 31.5 ± 3.8 pg/mL, both of which were significantly higher than the mean from control mice, 15.5 ± 3.8 pg/mL (Figure 6C, *p* < 0.01). Mice treated with free cisplatin showed a mean TNF-α value of 22.3 ± 4.2 pg/mL, which was not significantly different than control. The higher levels of TNF-α in the serum of mC-MVs and mE-MVs treated mice demonstrate an inflammatory response induced by MV treatment and suggest MVs may be able to induce a pro-inflammatory tumor microenvironment.

### 3.7. Human PBMC Derived M1-MV Uptake and Localization

Because cross-species differences can alter vesicle targeting and internalization, and to support future human clinical trials, the internalization of human PBMCs-derived hE-MVs uptake and localization in a human cancer cell line was examined. hE-MVs were added to the human OS cell line 143B culture for 24 h, then fixed and stained with the M1 marker CD86 labeled with secondary gold nanoparticles. hE-MVs were visualized inside 143B cells using IGSS and SEM (Figure 7). Gold-labeled hE-MVs, depicted as black dots (Figure 7 and Appendix A, red arrows), are clearly visualized inside the cell and in proximity to what is possibly a phagolysosome and mitochondria. The size of the internalized hE-MVs is approximately 150 nm (Figure 7, green bars). This experiment demonstrates the targeting and uptake ability of human PBMC-derived MVs into human OS cancer cells.

## 4. Discussion

Endogenous EVs are nano-sized, cargo-baring, lipid bilayer vesicles naturally released by cells into the extracellular space. The unique cellular targeting ability of EVs has led to the exploration as therapeutic agents [19]. Importantly, studies have shown that M1 macrophages actively secrete EVs that are able to repolarize M2 macrophages in the tumor microenvironment to the M1 or anti-cancer phenotype [12]. Despite the promise of EVs as therapeutic agents, the field is limited by the inability to produce EVs on a large scale. More recently, advances have been made in scaling up the production of MVDVs using cell disruption strategies. We recently demonstrated that M1-MVs can be generated using nitrogen cavitation of LPS- and IFN-γ-stimulated macrophages, and that 100 million M1-macrophages generated approximately two trillion hM1-MVs. Importantly, manufactured hM1-MVs retained both the tumor-targeting and pro-inflammatory cytokine phenotypes of naturally occurring EVs released by M1 macrophages [10].

In this work, we characterize human and mouse M1-MVs derived by nitrogen cavitation using scanning electron microscopy and show that they are similar in size and shape to naturally occurring extracellular vesicles, with rounded morphology and a size of approximately 100–200 nm in diameter [20]. Interestingly, we also show via CD86-labeled, enhancement staining transmission electron microscopy, the internalization and location of hM1-MVs within an OS cancer cell line. We demonstrate the ability of hE-MVs to physically enter human OS cells and localize to intracellular compartments consistent with phagolysosomes, supporting a plausible delivery method for cargo.

EVs have garnered considerable interest as therapeutic delivery vehicles to transport cargo across cell membranes because they have been shown to precisely target cell types via distinct surface molecules [21]. Previously, we demonstrated that MCDVs exhibit cell-specific targeting, preferentially homing to their cell of origin, and could easily be loaded to deliver cargo selectively to tumors in murine cancer models [12]. We demonstrate that mE-MVs home to tumors and also localize to the mouse liver. Systemically administered MCDVs have been shown to preferentially accumulate in the liver, and hepatic clearance is the dominant elimination route [22].

Here we also demonstrate that the incorporation of cisplatin into the MVs has superior anti-cancer efficacy against OS cell lines, as indicated by lower IC50 and compared with free cisplatin in vitro. C-MVs treatment resulted in DNA damage, indicating the cisplatin cargo, a DNA crosslinking agent, was effectively delivered inside the cell. In addition, the effectiveness of C-MVs was demonstrated in vivo using an orthotopic OS mouse model, with C-MV treatment resulting in a slower tumor growth rate and prolonged survival. Similarly, others have demonstrated the therapeutic effectiveness of chemotherapy-loaded endogenous M1 macrophage-derived exosomes in a breast cancer model [23]. However, we may be the first to successfully use manufactured, chemotherapy-loaded vesicles as anti-cancer agents. We previously demonstrated that both E-MVs and C-MVs were effective in an ovarian cancer mouse xenograft model, with both exhibiting significantly longer survival than control-treated mice [13]. In contrast, E-MV treatment was not superior to control in the current OS orthotopic model. This finding could be related to differences between the models, including tumor mutational burden, different mouse strains (nudes vs. BALBc/SCID), and the very rapid tumor growth rate of the OS model. Neither E-MVs nor C-MVs significantly reduced the formation of lung metastatic nodules, which may be attributed to the extended duration before euthanasia, allowing additional time for nodules to develop.

As cisplatin causes numerous off-target toxicities, often requiring patients to receive reduced doses or to discontinue therapy, toxicity was evaluated in mice treated with free cisplatin in comparison with equal doses of cisplatin incorporated into mM1-MVs. Mice treated with free cisplatin demonstrated significant weight loss, and although mC-MVs-treated mice did not increase in weight as fast as controls, none experienced weight loss. Cisplatin-treated mice also had significant elevations in serum BUN, ALP, and the albumin/globulin ratio and pathological changes indicative of both nephrotoxicity and hepatoxicity, which were not noted in mice treated with mE-MVs or mC-MVs treatment, suggesting mC-MVs are able to selectively deliver cisplatin to the tumor, reducing off-target toxicities.

In an orthotopic OS mouse model, we show that labeled M1-MVs selectively localize to the tumor xenograft and effectively increase the M1/M2 macrophage ratio within the tumor microenvironment, along with elevated plasma TNF-α levels. These findings suggest that M1-MVs can reprogram immunosuppressive M2 tumor-associated macrophages toward a pro-inflammatory M1 phenotype, potentially enhancing anti-tumor immune responses and improving the efficacy of PD-1/PD-L1 checkpoint inhibitors. The observed tumor selectivity, immunomodulatory activity, favorable safety profile, and capacity to reshape macrophage polarization support further investigation of M1-MVs as a promising therapeutic strategy for OS.

## 5. Conclusions

Nitrogen cavitation is an effective way of generating large-scale MVs that resemble naturally occurring EVs in cancer targeting ability. In an orthotopic OS model, mE-MVs and mC-MVs were both effective at increasing the pro-inflammatory tumor microenvironment, suggesting future therapeutic strategies in combination with immunotherapy. Finally, C-MVs are an innovative, more effective, and less toxic chemotherapy that warrants further development for the treatment of osteosarcoma.

## Figures and Tables

**Figure 1 cells-14-01616-f001:**
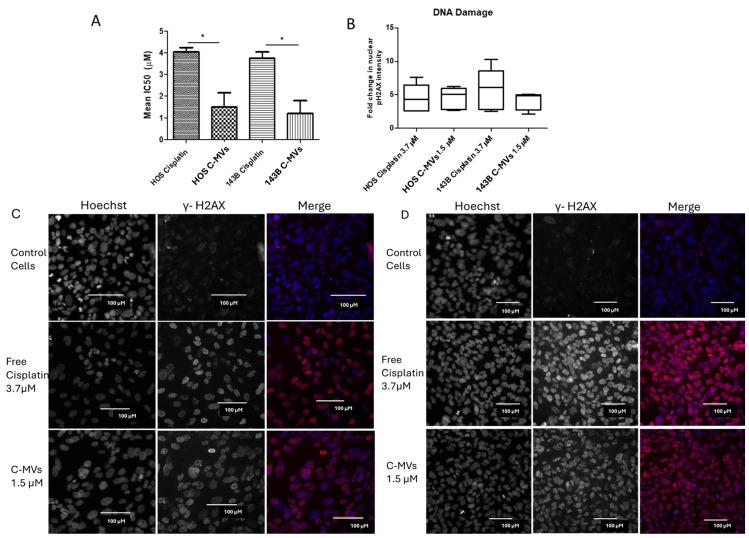
hC-MVs demonstrate enhanced OS cell toxicity and DNA damage efficiency compared to Free cisplatin. (**A**) Osteosarcoma cell line IC50 values ± SEM, estimated following 96 h treatment of increasing concentrations of hC-MVs or free cisplatin, (n = 3, * *p* < 0.05, Student’s unpaired *t*-test). (**B**) Quantification of nuclear pH2AX staining in HOS (**C**) and 143B cells (**D**). Mean fold change in nuclear fluorescent intensity signal when normalized to control cells ± SEM. (**C**) Representative images showing HOS cells and (**D**) 143B cells treated with control, free cisplatin, and hC-MVs at an IC50 value of 3.7 µM and 1.5 µM, respectively (n = 6, *p* > 0.05, one-way ANOVA).

**Figure 2 cells-14-01616-f002:**
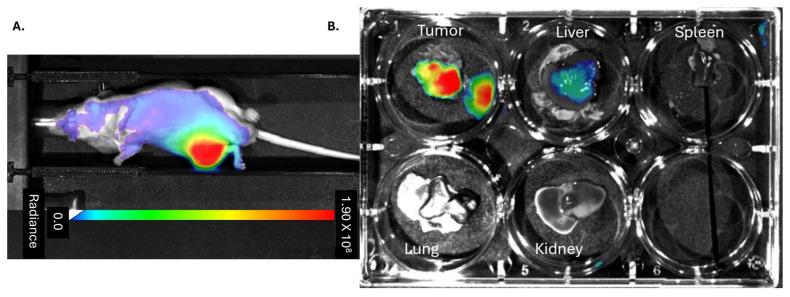
Localization of MVs. (**A**) Fluorescence image, excitation 710 and emission 770 nm of a mouse bearing 143B OS orthotopically in the right tibia. The mouse was injected with fluorescent DiR dye-labeled mE-MVs. (**B**) A necropsy of the mouse tumor, liver, spleen, kidney, and lung was performed. mE-MVs showed localization to the tumor and in the liver.

**Figure 3 cells-14-01616-f003:**
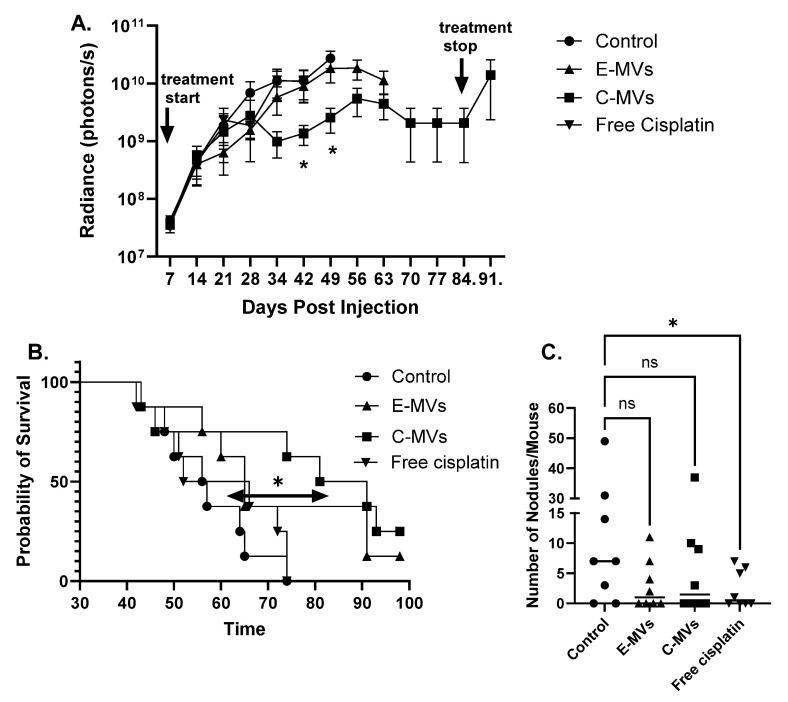
Anti-cancer activity of mC-MVs and mE-MVs in an OS xenograft mouse model. 143B luciferase-labeled cells were injected into the tibia of 32 mice. Mice were treated with Control (PBS), Free cisplatin (4.2 ± 0.6 mg/kg), mE-MVs or mC-MVs (6.56 × 10^11^ ± 9.04 × 10^10^ vesicles per injection) for up to 12 weeks. (**A**) Mean bioluminescence (photons per second) emitted from the tumor ± SEM. Data continues until the majority of the mice in the group are euthanized. mC-MVs showed significantly lower bioluminescence compared to control mice at 6 and 7 weeks of treatment (n = 8 per group, * *p* < 0.05, two-way ANOVA Fisher’s LSD). (**B**) Kaplan–Meier survival curve demonstrates a significantly longer survival in mice treated with mC-MVs compared to Control (*p* = 0.014, log-rank Mantel–Cox test). (**C**) In euthanasia, lung tissues from mice were harvested, and H&E staining was performed. Metastatic nodules were counted and plotted as the mean number ± SEM. Free cisplatin resulted in significantly fewer nodules than control (*p* = 0.047, ns = not significant, one-way ANOVA, Fisher’s LSD).

**Figure 4 cells-14-01616-f004:**
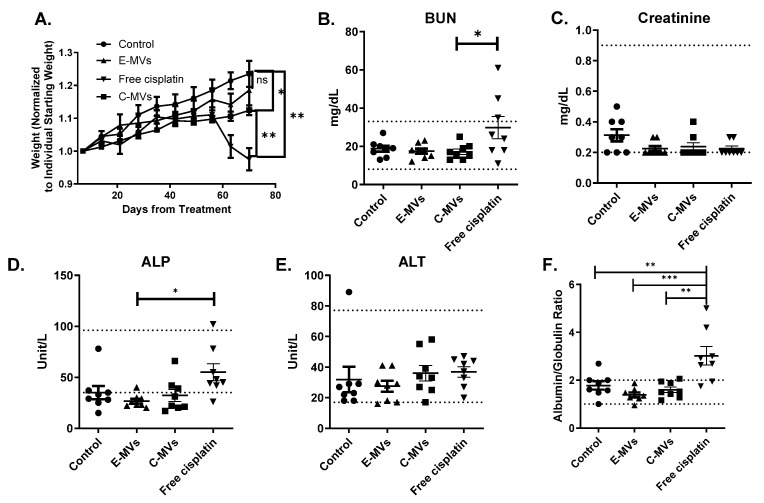
Free cisplatin treatment results in weight loss and abnormal serum chemistries. (**A**) Mice were weighed weekly. Weights were normalized to individual starting weights, then averaged within treatment groups and graphed ± SEM until fewer than 5 mice per group remained. (**B**) Mean serum BUN was significantly elevated in free cisplatin-treated mice vs. mE-MVs and mC-MVs. (**C**) Mean serum creatinine levels were not elevated in any mouse. (**D**) Mean serum ALP levels were significantly elevated in free cisplatin-treated mice when compared to mE-MVs-treated mice. (**E**) Mean serum levels of ALT were not significantly different between groups. (**F**) The albumin/globulin ratio was significantly elevated in mice treated with the free cisplatin group compared to the control, mE-MVs, and mC-MVs. (n = 32, 8 mice/group, * *p* < 0.05, ** *p* < 0.01, *** *p* < 0.001, ns = not significant one-way ANOVA with Fisher’s LSD).

**Figure 5 cells-14-01616-f005:**
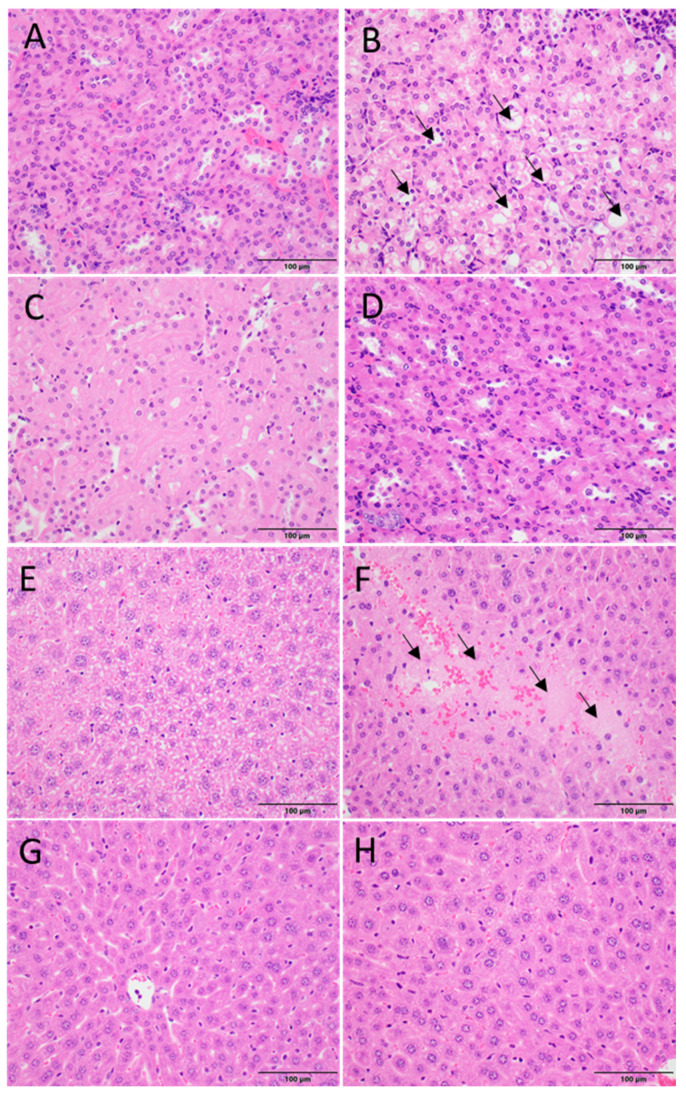
mE-MVs and mC-MVs treatment appear safe, while free cisplatin treatment induced nephrotoxicity and hepatotoxicity in mice. (**A**–**D**) Histopathological H&E stains of kidney sections of mice. (**A**) Control treated. (**B**) Free cisplatin-treated mice (black arrows show cytoplasmic vacuolization corresponding to a lack of mitochondria in type II renal tubular epithelial cells). (**C**) mE-MVs treated and (**D**) mC-MVs treated. (**E**–**H**) Histopathological H&E stain of liver sections. (**E**) Control treated. (**F**) Free cisplatin-treated mice (black arrows show multifocal areas of hepatocyte necrosis). (**G**) mE-MVs treated, and (**H**) mC-MVs treated. Bar = 100 µm.

**Figure 6 cells-14-01616-f006:**
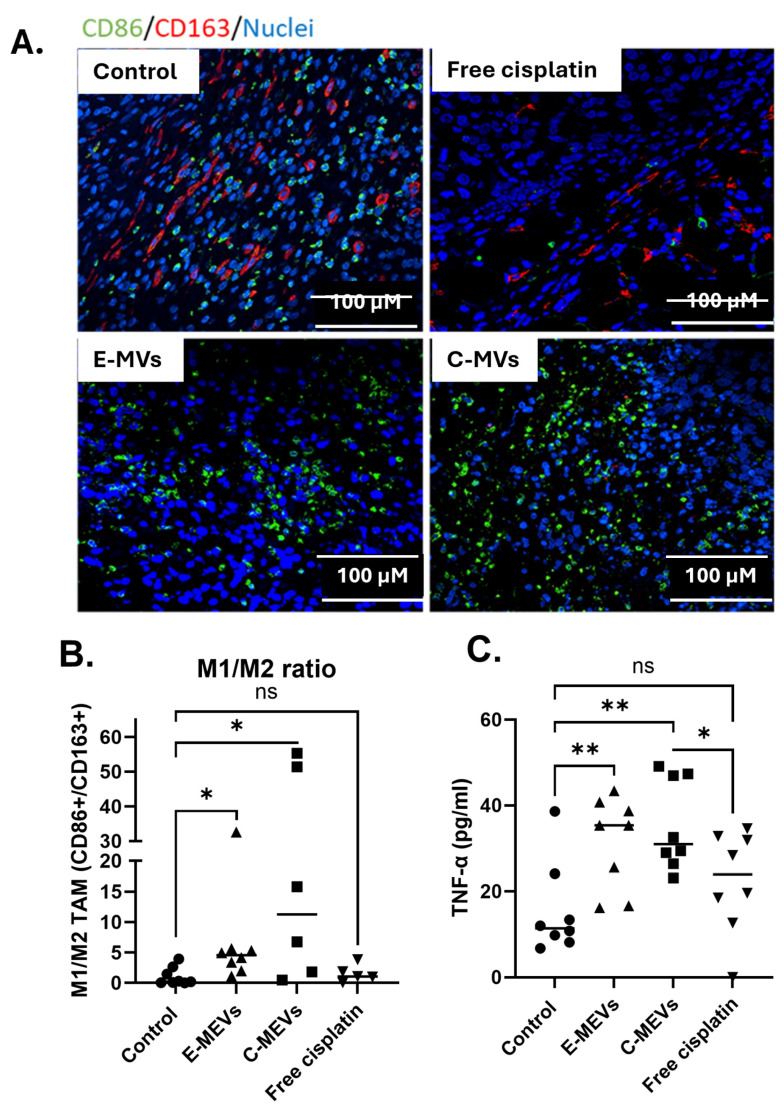
MVs treatment increases the M1/M2 Macrophage ratio in OS tibia xenografts and increases TNF-α. OS tibia xenograft tumors were harvested from mice treated with control, mE-MVs, mC-MVs, or free cisplatin, and immunohistochemical analysis for CD86+ (M1 macrophage marker, green stain), CD163+ (M2 macrophage marker, red stain), and DAPI (nuclei, blue stain) was performed. A 20X lens was used to observe all images, and the scale bar is approximately 100 µM. (**A**) Representative images of mice in each treatment group. (**B**) 5 random regions were selected, and the M1/M2 macrophage ratio was counted in tumors harvested from each mouse. Three mice from the free cisplatin group and two mice from the mC-MVs group did not have enough tumor to visualize. The ratio of M1/M2 stained cells was calculated from 5 random regions (n = 5–8, * *p* ≤ 0.05, ns = not significant, ANOVA Kruskal–Wallis test). (**C**) Serum from the same mice was assayed for TNF-α using ELISA, and the concentrations were calculated. One mouse each from control and free cisplatin had concentration below the limit of quantification, and the values were extrapolated from the standard curve (n = 7–8, * *p* ≤ 0.05, ** *p* < 0.01, ns = not significant, one-way ANOVA Fisher’s LSD).

**Figure 7 cells-14-01616-f007:**
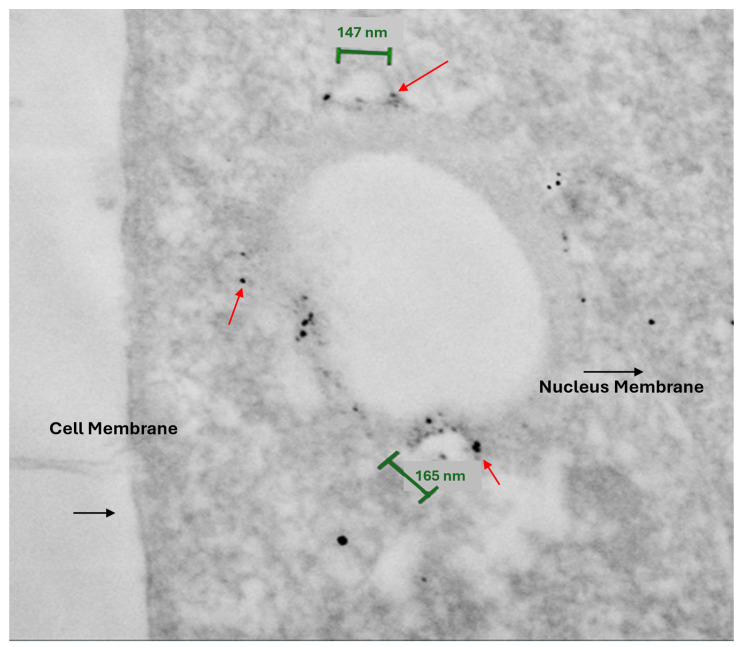
Transmission electron microscopy of hMVs internalized in a 143B cell. hMVs labeled with gold nanoparticles (tiny black dots shown with red arrows). hMVs measure 165.4 and 146.8 nm (shown with green bars).

## Data Availability

Data available is within the article or its Appendix A and available upon request.

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
