# Peer review of "Cisplatin-Loaded M1 Macrophage-Derived Vesicles Have Anti-Cancer Activity in Osteosarcoma"

_cells, 2025, doi:10.3390/cells14201616_

Round 1

Reviewer 1 Report

Comments and Suggestions for Authors

The authors have presented a potentially interesting and relevant study in the field of oncology, in which C-MVs are a promising therapeutic strategy for osteosarcoma. However, in its current form the manuscript contains numerous formal and structural errors significantly hinder its reading and scientific evaluation. Unfortunately  the manuscript has been submitted with little attention to detail.

In particular, I would like to highlight the following main concerns:

-Figures 1C and D are of low resolution and the fluorescence signals are not clearly visible. Scale bars are missing and the abbreviations should be consistent with those used in the image and legend.

-Figure 6A needs a scale bar.

-The graphical abstract is missing

-More TEM images are needed to evaluate the uptake and localization of immunogold-labeled CD86.

-Figure legends are not organized consistently and in some cases lack clarity (see Figure 5).

-Comments on the results are only provided for some assays. In any case, such interpretations should be included in the Discussion section, not at the end of the Results section.

-The aim of the experiments is not always clearly stated, making it difficult to understand the rationale behind them and to interpret the data correctly.

-The rationale and novelty of the current study should be stated more clearly to highlight its contribution.

-In the introduction, the role of macrophage polarization needs to be established more explicitly.

-Both the Introduction and Discussion sections require further elaboration to provide adequate context and scientific interpretation

For these reasons, the manuscript cannot be accepted for publication in its current form. I recommend that the authors undertake a thorough revision of both the formal aspects and the scientific content to significantly improve the overall quality of the work.

Author Response

Comments and Suggestions for Authors

The authors have presented a potentially interesting and relevant study in the field of oncology, in which C-MVs are a promising therapeutic strategy for osteosarcoma. However, in its current form the manuscript contains numerous formal and structural errors significantly hinder its reading and scientific evaluation. Unfortunately  the manuscript has been submitted with little attention to detail.

In particular, I would like to highlight the following main concerns:

-Figures 1C and D are of low resolution and the fluorescence signals are not clearly visible. Scale bars are missing and the abbreviations should be consistent with those used in the image and legend.

We have edited Figure 1 to include higher resolution and scale bars.  We have edited the abbreviations for consistency. 

-Figure 6A needs a scale bar.

A 20 X lens was used to capture all images, and a scale bar has been more clearly defined in the free cisplatin group.

-The graphical abstract is missing

I am sorry for the oversite. We have now included the graphical abstract.

-More TEM images are needed to evaluate the uptake and localization of immunogold-labeled CD86.

We have added two more images as Supplemental Figure 5 A,B.  

-Figure legends are not organized consistently and in some cases lack clarity (see Figure 5).

Figure legends have been edited for clarity and consistency.

-Comments on the results are only provided for some assays. In any case, such interpretations should be included in the Discussion section, not at the end of the Results section.

Thank you for your comment. We have added an interpretation of each of the experiments to the discussion (see highlighted area).

-The aim of the experiments is not always clearly stated, making it difficult to understand the rationale behind them and to interpret the data correctly.

The rationale behind the experiment has now been clearly stated in each results section (see highlighted area).

-The rationale and novelty of the current study should be stated more clearly to highlight its contribution.

We have added lines to the discussion about rationale and novelty of the study (see highlighted area).

-In the introduction, the role of macrophage polarization needs to be established more explicitly.

We have expanded the paragraph on macrophage polarization in the introduction (see highlighted area).

-Both the Introduction and Discussion sections require further elaboration to provide adequate context and scientific interpretation

We have expanded both the introduction and discussion sections (see highlighted areas).

For these reasons, the manuscript cannot be accepted for publication in its current form. I recommend that the authors undertake a thorough revision of both the formal aspects and the scientific content to significantly improve the overall quality of the work.

Reviewer 2 Report

Comments and Suggestions for Authors

Review of the MS cells-3892018, titled: Cisplatin Loaded M1 Macrophage Derived Vesicles Have Anti-Cancer Activity in Osteosarcoma.

This paper presents an interesting approach for osteosarcoma therapy using cisplatin-loaded M1 macrophage-derived vesicles (C-MVs). The concept utilizes macrophage tropism for tumors to enhance drug delivery while minimizing toxicity, a strategy with significant potential application. The methodology is rigorous, and results are robust.

C-MVs demonstrate superior tumor targeting and reduced systemic toxicity compared to free cisplatin, addressing a critical clinical challenge for osteosarcoma. For the modulation of the tumor immune microenvironment, the authors developed manufactured M1 macrophage‑derived vesicles (MVs), loaded them with cisplatin (C‑MVs), and compared empty MVs (E‑MVs), free cisplatin, and control in vitro and in an orthotopic 143B tibial xenograft model. Authors  found that  C‑MVs show lower IC50 in HOS and 143B cell lines than free cisplatin while causing equivalent DNA damage per measured endpoint; Also,  MV localize to orthotopic tumors after IP administration;  mC‑MVs slow tumor growth and prolong survival compared to control, with less systemic toxicity than equivalent free cisplatin; and finally, both E‑MVs and C‑MVs shift tumor macrophage populations toward an M1 phenotype (increased M1/M2 ratio) and raise serum TNF‑α.  Furthermore, orthotopic xenografts with bioluminescence monitoring provide robust in vivo validation. Furthermore, rigorous evaluation of organ damage (liver/kidney histopathology, serum chemistry) highlights C-MVs  safety advantages.

Minor Revisions

  • Unit Formatting:

Several temperatures and centrifugation steps are inconsistently reported (e.g., “370 C”, “40 C”, “4º C”); correct these (use 37 °C, 4 °C, etc.) and standardize centrifugation notation (e.g., 10,000 × g, 100,000 × g).

(5 X 105" → "5 × 10^5"). Provide a consistent style (units, spacing, p‑values, n formatting).

  • Figure Legends:

Define all abbreviations (e.g., E-MVs, C-MVs) in legends; clarify statistical annotations (e.g., *p < 0.05).

 Please provide analytical validation for cisplatin quantification (LLOQ, ULOQ, intra/inter‑assay CVs) or cite validated methods.

L 224-225 : remove space enter.

- pH2AX images: could you  include quantification per cell and representative single‑cell images. Also,  include scale bars.

Author Response

Unit Formatting:

Several temperatures and centrifugation steps are inconsistently reported (e.g., “370 C”, “40 C”, “4º C”); correct these (use 37 °C, 4 °C, etc.) and standardize centrifugation notation (e.g., 10,000 × g, 100,000 × g).

(5 X 105" → "5 × 10^5"). Provide a consistent style (units, spacing, p‑values, n formatting).

Thank you for your edits. Unfortunately, many of these errors occurred when the journal formatted the text for review. We have fixed the errors and made the figure legends style consistent.

Figure Legends:

Define all abbreviations (e.g., E-MVs, C-MVs) in legends; clarify statistical annotations (e.g., *p < 0.05).

Thank you for your careful review.  We have made the abbreviations and statistical annotations consistent.

 Please provide analytical validation for cisplatin quantification (LLOQ, ULOQ, intra/inter‑assay CVs) or cite validated methods.

The analytical validation for cisplatin quantification was performed according to Agilent Technologies white pages https://www.agilent.com/cs/library/applications/5991-9189EN_cisplatin_determination_LCMSMS_application.pdf

This is now reference 15.

L 224-225 : remove space enter.

Removed.

- pH2AX images: could you  include quantification per cell and representative single‑cell images. Also,  include scale bars.

Single cell images were not possible with the resolution of the fluorescent camara that we used.  We have put scale bars on the image. 

Round 2

Reviewer 1 Report

Comments and Suggestions for Authors

I would like to thank the author for taking my suggestions on board, as I believe this has made the manuscript both more complete and more interesting. It can be accepted in its current form.